



**Basic chemical compositions combination rules and quantitative criterion of red beds**
Guangjun Cui [1,2], Jin Liao [2], Linghua Kong [2], Cuiying Zhou [2,*], Zhen Liu [2,*], Lei Yu [2],
Lihai Zhang [3]
[1] Institute of Estuarine and Coastal Research/Guangdong Provincial Engineering Research Center of
Coasts, Islands and Reefs, School of Ocean Engineering and Technology, Sun Yat-sen University,
Guangzhou 510275, China
[2] Guangdong Engineering Research Center for Major Infrastructures Safety, Sun Yat-sen University,
Guangzhou, 510275, China
[3] The University of Melbourne, Melbourne VIC 3010, Australia
*Correspondences: zhoucy@mail.sysu.edu.cn (C. Zhou), liuzh8@mail.sysu.edu.cn (Z. Liu)

## Abstract

Red beds belong to slippery formations, and their rapid identification is of great significance for
major scientific and engineering issues such as geological hazard risk assessment and rapid response.
Existing research often identifies red beds from a qualitative or semi quantitative perspective, resulting
in slow recognition speed and inaccurate recognition results, making it difficult to quickly handle
landslide geological disasters. Combined with the correlation between red beds geomorphic
characteristics, mineral compositions, and chemical compositions, this study established a rapid
quantitative identification criterion based on the basic chemical compositions combination rules in the
red beds. By collecting chemical compositions data of rocks containing red beds, a total of 241,405
groups data were collected for qualitative and quantitative comparison between multiple sets of
chemical composition combinations. The results indicate that simultaneously meeting the following
chemical composition combinations can serve as a quantitative criterion for distinguishing red beds
from other rocks: $SiO_2+Al_2O_3 \approx 50.7\%\sim85.0\%$, $Al_2O_3/SiO_2 \approx 0.14\sim0.41$, $FeO+Fe_2O_3 \approx 0.9\%\sim7.9\%$,
$Fe_2O_3/FeO \approx 1.52\sim7.70$,  $K_2O+Na_2O \approx 1.6\%\sim6.8\%$,  $Na_2O/K_2O \approx 0.02\sim0.43$,



CaO+MgO ≈ 0.8%~9.2%. By comparing the chemical composition combinations of 15 kinds of rocks
collected from China in this study, it is proven that the quantitative criterion proposed in this study are
effective.
**Keywords:** red beds, quantitative criterion, geological disasters, rapid response, chemical compositions

**1. Introduction**

Red beds are widely distributed throughout the world (Chen et al. 2021; Yan et al. 2019; Zhou et

al. 2023). Geological disasters occur frequently in the red beds distribution area, especially landslides,
debris flows, and collapses (Chen et al. 2014). According to the characteristics of disasters such as
landslides, the red beds belong to "landslide prone strata", and the instability of slopes with weak
interlayers of the red beds is particularly evident (Zhang et al. 2015). This is mainly due to the strong
hydrophilicity and weak permeability of the red beds, which are prone to softening and plastic
deformation under the action of water; After absorbing water, the red beds are easy to expand, and after
losing water, they are easy to contract; The weathering resistance of the red beds are weak, they are
easy to collapse, and their compressive and shear strength are low (Marat et al. 2022; Wang et al. 2017;
Wu et al. 2018; Zhang et al. 2016). The red beds have different lithology or poor binding force with
other rock strata, which can easily cause differential deformation and lead to rock mass sliding along
the bedding plane (He et al. 2023; Liu et al. 2020). Therefore, the identification of rock types, especially
the rapid determination of red beds, is of great significance for major scientific and engineering issues
such as risk assessment and rapid response of geological disasters in red beds distribution area.

At present, the studies on red beds identification are mostly carried out from the perspectives of

geomorphic characteristics, mineral compositions, and chemical compositions (Cui et al. 2022; Zhou et
al. 2021). Among them, the research of geomorphic characteristics and mineral compositions mostly
adopts qualitative or semi quantitative methods, and there are many such studies. For example, Rainoldi
et al. (2015) identified red beds by studying the color of geomorphic characteristics and hematite in
mineral compositions, and studied the mechanism of red beds bleaching. Uchida et al. (2000)
distinguished red sandstone, yellowish brown sandstone, and green sandstone according to the content



of hematite, goethite, biotite, and muscovite in the mineral compositions, analyzed the characteristics
of different rocks and pointedly protected Angkor monuments. Xue et al. (2023) distinguished red
mudstone and red sandstone by quantifying the clay mineral content in the mineral compositions, in
order to analyze the mechanisms and control factors of summer uplift of high-speed railway cutting. In
addition, some scholars have conducted quantitative studies on the chemical compositions of red beds,
and such studies are less. Hong et al. (2009) analyzed the alteration of clay minerals by studying the
changes in the $SiO_2/Al_2O_3$ ratio in the chemical compositions of the red beds, thereby obtaining the
weathering degree of the red beds. Bankole et al. (2016) studied the relationship between Fe/Mg ratio,
$Fe^{3+}/FeT$ ratio, and Cr/Fe ratio of red beds to indirectly study the oxygen content of the Paleoproterozoic.
Hu et al. (2006) studied the characteristics of high $Fe_2O_3$ content and low FeO content in the oceanic
red beds, and analyzed ancient landslides on the continental margin from the perspective of petrology.
However, these studies do not distinguish between red beds and other rocks in terms of chemical
compositions. The use of portable spectrometers and drone-borne multi-sensor remote sensing
technique can quickly obtain the chemical compositions of rocks in geological disasters while ensuring
safety (Kirsch et al. 2018; Triantafyllou et al. 2021), making it feasible to use chemical compositions
as the standards to distinguish red beds from other rocks.

Therefore, the purpose of this study to develop a quantitative criterion for quickly and accurately

identifying the red beds. Figure 1 shows the methodology used in this study involving the investigation
of geomorphic characteristics, mineral compositions, and chemical compositions. There are few studies
on identifying red beds from the perspective of chemical compositions, which is the focus of this study.
Moreover, there is a close relationship between geomorphic characteristics, mineral compositions, and
chemical compositions (Moonjun et al. 2017). This study first collected the data about the geomorphic
characteristics, mineral content, and chemical composition of red beds and other rocks, then compared
these data to obtain the basic characteristics of red beds, and finally summarized and analyzed the red
beds identification criterion and verified the reliability of this criterion.



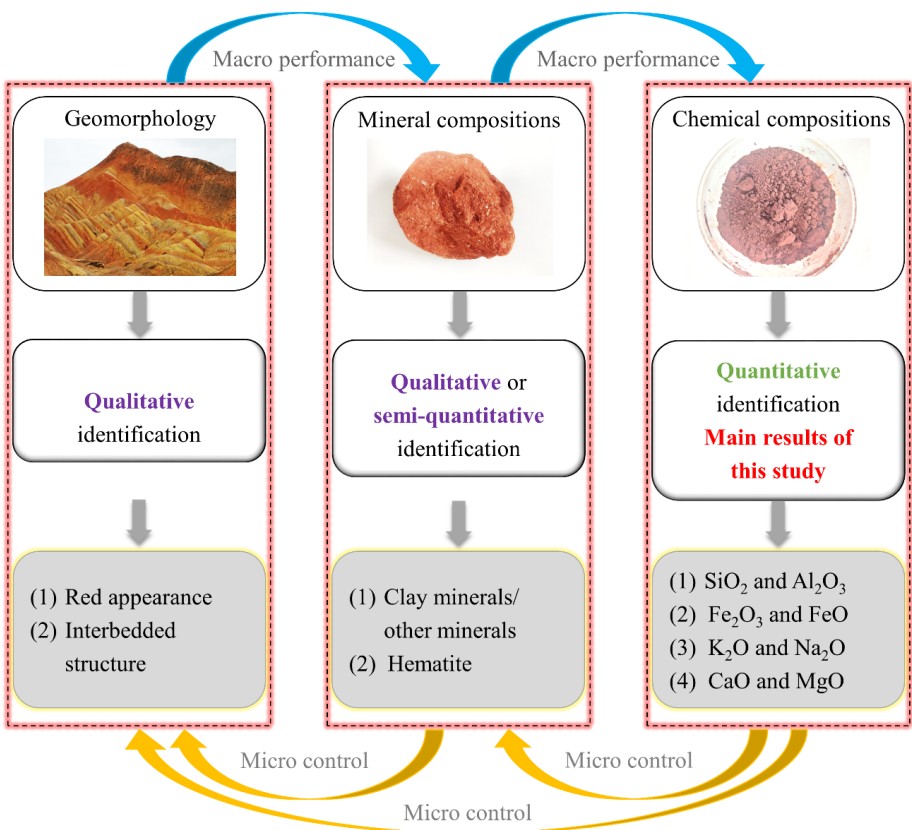


**Figure 1.** Methodology proposed in study for identifying red beds from geomorphic characteristics, mineral compositions, and chemical compositions.


**2. Methods**
2.1 Data collection
A large amount of data for red beds and other rocks was collected from previous studies and
analysed from geomorphic characteristics, mineral compositions, and chemical compositions
perspectives. Data were collected from the previous studies about landslides, debris flows, and collapses
on the geomorphic characteristics of red beds, igneous rocks (andesite, basalt, diorite, granite),
metamorphic rocks (gneiss, marble), and other sedimentary rocks (arkose, black-shale, breccia,
claystone, dolomite, lignite, limestone, marl, mudstone, siliciclastic, tuff) (e.g., (Anbarasu et al. 2010;
Ciftci et al. 2008; Contino et al. 2017; de Montety et al. 2007; Gokbulak and Ozcan 2008; Hale et al.



2021; Harp et al. 2011; He et al. 2021; Kavvadas et al. 2020; Li et al. 2016; Liu et al. 2018; Ni et al.
2015; Perez-Rey et al. 2019; San et al. 2020; Underwood et al. 2016; Wang et al. 2022; Xia et al. 2019;
Zhang et al. 2015; Zhang et al. 2017)). The geomorphic characteristics of red beds investigated in this
study involve the evolution process and distribution of red beds on Earth's surface, and the results were
compared with that of other types of rock samples.
Previous studies have shown that there is a relationship between mineral compositions and
geomorphic characteristics of red beds during the geological processes (Bankole et al. 2016). This study
mainly focuses on the influence of mineral compositions on geomorphic characteristics, particularly the
layered structure and color of red beds. The mineral compositions of red beds (1,536 groups data) were
collected from the previous studies as shown in Supplementary Table 1 (e.g., (Bai et al. 2020; Chen et
al. 2014; Jian et al. 2009; Li et al. 2023; Li et al. 2015; Li et al. 2013; Liu et al. 2020; Marat et al. 2022;
Wang et al. 2018; Wang et al. 2014; Wang et al. 2017; Yao et al. 2016; Zha et al. 2022; Zhang et al.
2016; Zhang et al. 2020; Zhang et al. 2021)). These studies used semi quantitative or quantitative
methods in XRD technology to statistically analyze the differences in mineral composition between
different red beds (*e.g.,* quartz, feldspar, mica, hematite, clay minerals, and calcite), as detailed in the
aforementioned literatures.
Moreover, previous studies have shown that the geomorphic characteristics and mineral
compositions of rocks are strongly correlated to their chemical compositions (Perri et al. 2013). For
example, the content of $Fe_2O_3$ or hematite in the red beds is higher than that in the grey beds (Hu et al.
2006). The chemical compositions of red beds (1536 groups data) with different geological ages and
various lithologies such as conglomerate, sandy conglomerate, sandstone, siltstone, shale and mudstone
were collected from the previous studies as shown in Supplementary Table 2 (e.g., (Gao et al. 2017;
Hong et al. 2009; Jiang et al. 2022; Kong et al. 2018; Liu et al. 2007; Liu et al. 2020; Liu et al. 2006;
Uchida et al. 2000; Wild et al. 2017; Xue et al. 2023; Yang et al. 2016; Zhang et al. 2008; Zhao et al.
2005; Zhu et al. 2003)). The chemical compositions of igneous rocks, including andesite
(Supplementary Table 3 - 49,203 groups data. Data were downloaded from the GEOROC database
(https://georoc.mpch-mainz.gwdg.de//georoc/) on 11 May 2023, using the following parameters: search
= andesite.), basalt (Supplementary Table 4 - 80,365 groups data. Data were downloaded from the





GEOROC database on 11 May 2023, using the following parameters: search = basalt.), diorite
(Supplementary Table 5 - 4,941 groups data. Data were downloaded from the GEOROC database on
11 May 2023, using the following parameters: search = diorite.), and granite (Supplementary Table 6 -
17,272 groups data. Data were downloaded from the GEOROC database on 11 May 2023, using the
following parameters: search = granite.). The chemical compositions of metamorphic rocks, including
gneiss (Supplementary Table 7 - 24,300 groups data. The data were downloaded from the EarthChem
Portal Database (http://portal.earthchem.org/) on 20 April, 2018, using the following parameters:
material = metamorphic and rock name = gneiss.) and marble (Supplementary Table 8 - 3,364 groups
data. The data were downloaded from the EarthChem Portal Database on 12 May, 2023, using the
following parameters: material = metamorphic and rock name = marble.). The chemical compositions
of other sedimentary rocks, including arkose (Supplementary Table 9 - 682 groups data. The data were
downloaded from the EarthChem Portal Database on 10 May, 2023, using the following parameters:
material = sedimentary and rock name = arkose.), black-shale (Supplementary Table 10 - 305 groups
data. The data were downloaded from the EarthChem Portal Database on 10 May, 2023, using the
following parameters: material = sedimentary and rock name = black-shale.), breccia (Supplementary
Table 11 - 1,396 groups data. The data were downloaded from the EarthChem Portal Database on 10
May, 2023, using the following parameters: material = sedimentary and rock name = breccia.),
claystone (Supplementary Table 12 - 3,790 groups data. The data were downloaded from the
EarthChem Portal Database on 10 May, 2023, using the following parameters: material = sedimentary
and rock name = claystone.), dolomite (Supplementary Table 13 - 2,169 groups data. The data were
downloaded from the EarthChem Portal Database on 6 May, 2023, using the following parameters:
material = sedimentary and rock name = dolomite.), lignite (Supplementary Table 14 - 3 groups data.
The data were downloaded from the EarthChem Portal Database on 24 April, 2018, using the following
parameters: material = sedimentary and rock name = lignite.), limestone (Supplementary Table 15 -
9,104 groups data. The data were downloaded from the EarthChem Portal Database on 10 May, 2023,
using the following parameters: material = sedimentary and rock name = limestone.), marl
(Supplementary Table 16 - 142 groups data. The data were downloaded from the EarthChem Portal
Database on 10 May, 2023, using the following parameters: material = sedimentary and rock name =



marlstone, marl.), mudstone (Supplementary Table 17 - 6,140 groups data. The data were downloaded
from the EarthChem Portal Database on 10 May, 2023, using the following parameters: material =
sedimentary and rock name = mudstone, mud.), siliciclastic (Supplementary Table 18 - 26,938 groups
data. The data were downloaded from the EarthChem Portal Database on 10 May, 2023, using the
following parameters: material = sedimentary and rock name = siliciclastic.), tuff (Supplementary Table
19 - 10,295 groups data. The data were downloaded from the EarthChem Portal Database on 6 May,
2023, using the following parameters: material = sedimentary and rock name = tuff.). Due to the high
content of quartz, clay minerals, hematite, calcite, dolomite, feldspar, etc. in the red beds, the main
oxide components are $SiO_2$, $Al_2O_3$, $Fe_2O_3$, FeO, CaO, MgO, $Na_2O$, and $K_2O$, this study mainly focuses
on the differences in chemical compositions combination rules between the red beds and other rocks,
such as $SiO_2$ and $Al_2O_3$, $Fe_2O_3$ and FeO, CaO and MgO, $Na_2O$ and $K_2O$.

2.2 Criterion verification
In order to verify the proposed basic chemical compositions combination rules and quantitative
criterion of red beds, 15 kinds of rocks of known rock types were selected in Guangdong, Sichuan,
Hubei, Zhejiang, and Anhui provinces (Figure 2), including 12 kinds of red beds (red claystone, red
mudtone, red silty mudstone, red argillaceous siltstone, red fine sandstone, red medium sandstone, red
coarse sandstone, red conglomerate, etc.), limestone (1 kind), arkose (1 kind) and mudstone (1 kind).

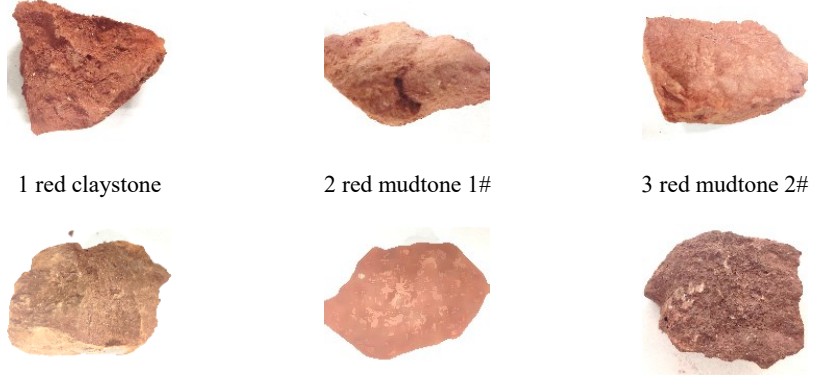

1 red claystone        2 red mudtone 1#        3 red mudtone 2#

4 red silty mudstone    5 red argillaceous siltstone 1#    6 red argillaceous siltstone 2#



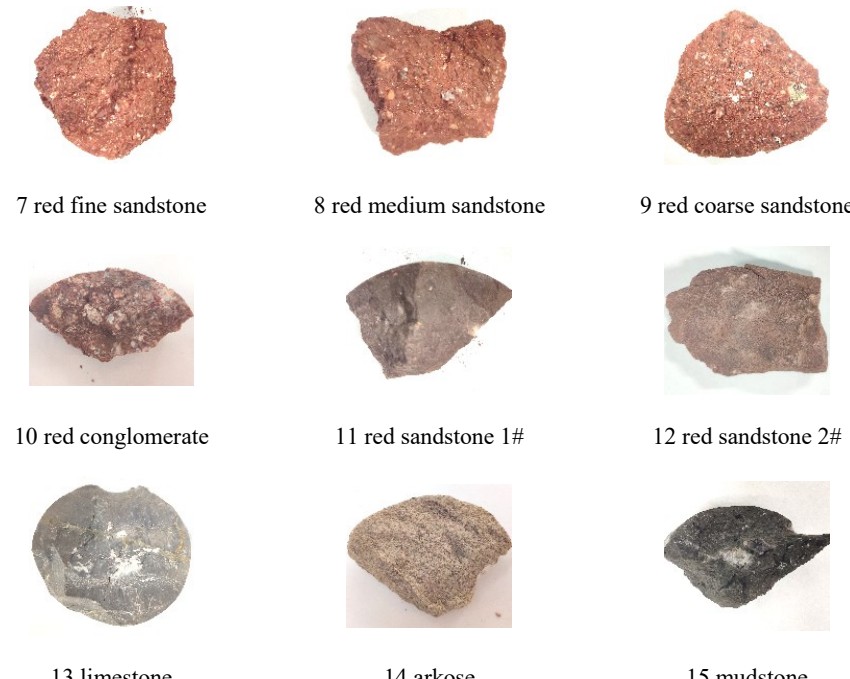

**Figure 2.** 15 kinds of rocks collected for the verification of the quantitative criterion.

These rock samples were analyzed by the MiX5 Pro handheld X-ray fluorescence element analyzer (Figure 3) of Sun Yat-sen University to check whether these elements conform to the basic chemical compositions combination rules of red beds proposed by this study. The working principle of this instrument is that a miniature X-ray source provides tube voltage and tube current, and the light tube emits continuous X-ray spectral lines. The X-rays irradiated on the sample generate X-ray fluorescence with sample characteristics, which is converted into voltage signals through detectors. On the instrument analysis interface, point the detection window towards the rock sample and press the trigger to start and stop the measurement. After amplification and data collection, the signal is processed to obtain the required test data. The instrument can detect elements with atomic number greater than or equal to 12, that is, element Na that cannot get the above attention (atomic number is 11). Therefore, the content of Na element is determined based on the median of $Na_2O/K_2O$ of the corresponding rock in Section 3.3 and K element detected by the MiX5 Pro handheld X-ray fluorescence element analyzer. Moreover, the Fe element content obtained by this instrument is the content of $Fe_2O_3+FeO$. The





corresponding $Fe_2O_3$ and FeO contents are determined based on the median of $Fe_2O_3$/FeO of the
corresponding rock in Section 3.3 and $Fe_2O_3$+FeO detected by the MiX5 Pro handheld X-ray
fluorescence element analyzer.

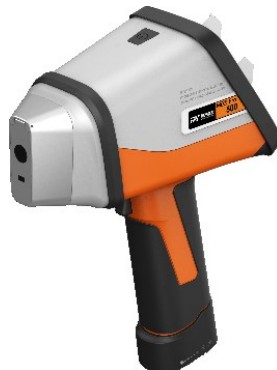


**Figure 3.** MiX5 Pro handheld X-ray fluorescence element analyzer.


**3. Results and discussions**
3.1 Geomorphic characteristics of red beds

Red beds are sedimentary rocks of different geological ages (mainly Mesozoic and Cenozoic) with

bedding structure typically consisting of various lithologies such as conglomerate, sandy conglomerate,
sandstone, siltstone, shale and mudstone that are predominantly red in color due to the presence of ferric
oxides (Yan et al. 2019). Owing to differences in depositional environments and influences of late stage
geologic processes, the color of red beds can be brownish-reddish-yellow, brownish-yellow, purplish-
red, brownish-red, grayish-purple and other reddish tints (Nance 2015; Yan et al. 2019), making it
difficult to accurately describe using the CIELAB color space and/or Munsell color system. Bedding is
a common structural feature of sedimentary rocks representing the changes in the sedimentary
environment. The sandstone is one of the most common types of red beds, with a distinct reddish
appearance. Compared with the red beds geomorphic characteristics, igneous rocks and metamorphic
rocks do not show the two characteristics of red appearance and bedding at the same time. Basalts are
reddish in appearance but does not have bedding (Cunha et al. 2005). In addition, andesites are mainly
light black and have a columnar structure which is similar to that of basalts (Feizizadeh et al. 2021).



Most of granites are grey or light brown with a significantly different structure compared to red beds
(Migon et al. 2018), while gneisses are generally characterized as a dark and light gneissic structure
(Garajeh et al. 2022). Although the red color appearance and bedding structure can be used as qualitative
criteria for identifying the red beds, the analysis of mineral and chemical compositions is still necessary
for identifying the rocks from quantitative perspective.

3.2 Mineral compositions of red beds
Table 1 shows the statistical analysis results of mineral compositions of red beds in Supplementary
Table 1. The common minerals in the red bed are quartz (median value is 40%, the same below), clay
minerals (35%, including kaolinite, illite, montmorillonite, and chlorite), feldspar (10%, including K-
feldspar and plagioclase), calcite (10%), mica (7%, including biotite, muscovite and sericite), and
hematite (3%) according to their content. According to the average value and standard deviation, it can
be seen that the content range of various minerals has significant dispersion. The ratio of the content of
clay minerals to other minerals (quartz, feldspar, mica, hematite, and calcite) ranges between 0.11 to
1.50. The hematite content ranges between 1.5% and 10.0% (percentile=10%~90%), and reddish
appearance of red beds is due to the abundant hematite content of the rocks. The change in mineral
compositions of red beds could lead to the change in rock color which is one of the major characteristics
of red beds. Furthermore, when the red beds encounter water, softening and expansion could happen
because of the large amount of clay minerals in the rocks, especially the mudstone. The differences in
mineral compositions of the red beds can also be quantitatively described through their chemical
composition combination characteristics (Table 2).
**Table 1.** Mineral compositions of red beds.

| Minerals | Range (per=0%~100%) | Range (per=10%~90%) | Median value (per=50%) | Average value | Standard deviation |
|---|---|---|---|---|---|
| Quartz (%) | 2.3~94.0 | 21.0~69.0 | 40.0 | 42.6 | 18.8 |
| Clay minerals (%) | 1.0~80.0 | 7.8~59.0 | 35.0 | 34.1 | 18.6 |
| Feldspar (%) | 0.4~71.0 | 2.3~25.0 | 10.0 | 12.6 | 10.7 |
| Mica (%) | 0.1~40.8 | 3.0~20.0 | 7.0 | 9.2 | 8.2 |



| | | | | |
|---|---|---|---|---|
| Hematite (%) | 0.4~25.2 | 1.5~10.0 | 3.0 | 5.0 | 4.4 |
| Calcite (%) | 0.7~97.7 | 3.1~23.5 | 10.0 | 12.2 | 10.0 |
| Clay minerals/ Other minerals | 0.01~6.00 | 0.11~1.50 | 0.61 | 0.76 | 0.66 |

Note: per – percentile; Other minerals – quartz, feldspar, mica, hematite, and calcite.

**Table 2.** Chemical composition of minerals in red beds (%).

| Mineral chemical formulas | $SiO_2$ | $Al_2O_3$ | $Fe_2O_3$ | FeO | CaO | MgO | $Na_2O$ | $K_2O$ | $H_2O$ | $CO_2$ |
|---|---|---|---|---|---|---|---|---|---|---|
| Quartz ($SiO_2$) | 100.0 | | | | | | | | | |
| Potassium feldspar ($KAlSi_3O_8$) | 64.7 | 18.4 | | | | | | 16.9 | | |
| Sodium feldspar ($NaAlSi_3O_8$) | 68.8 | 19.4 | | | | | 11.8 | | | |
| Calcium feldspar ($CaAl2Si_2O_8$) | 43.2 | 36.7 | | | 20.1 | | | | | |
| White mica ($KAl_2(AlSi_3O_{10})(OH,F)_2$) | 45.2 | 38.4 | | | | | | 11.8 | 4.1 | |
| Biotite ($KMg_3[Si_3 AlO_{10}](OH,F)_2$) | 43.0 | 12.2 | | | | 28.8 | | 11.2 | 2.2 | |
| Phlogopite ($K(Mg,Fe)_3AlSi_3O_{10}(F,OH)_2$) | 41.6 | 11.8 | | 8.3 | | 23.2 | 0.5 | 10.9 | 3.6 | |
| Hematite ($Fe_2O_3$) | | | 100.0 | | | | | | | |
| Calcite ($CaCO_3$) | | | | | 56.0 | | | | | 44.0 |
| Kaolinite ($Al_2Si_2O_5(OH)_4$) | 46.6 | 39.5 | | | | | | | 14.0 | |
| Illite ($K_{0.75}(Al_{1.75}R)[Si_{3.5}Al_{0.5}O_{10}](OH)_2$) | 54.0 | 17.0 | 1.9 | | | 3.1 | | 7.3 | 12.0 | |
| Montmorillonite ($(Na,Ca)_{0.33}(Al,Mg)_2[Si4O_{10}](OH)_2·nH_2O$) | 43.8 | 18.6 | | | 1.0 | | 1.1 | | 36.1 | |
| Chlorite ($Y_3[Z_4O_{10}](OH)_2·Y_3(OH)_6$) | 30.3 | 17.1 | | 15.1 | | 25.4 | | | 12.1 | |

Note: Data collected from http://webmineral.com/ and https://www.mindat.org/.

3.3 Chemical composition characteristics of red beds

Figures 4~5 are mainly used to qualitatively analyze the differences in chemical compositions between the red beds and other rocks through scatter plots. The area surrounded by black dashed lines is the area where the red beds data points are located. To better distinguish various rock data points, the distribution areas of various rock data are shown on the right side of the figure, and the corresponding colored dashed ellipses are used to indicate the distribution areas in the dataset. Figure 4 shows the comparison of $SiO_2$ and $Al_2O_3$, FeO and $Fe_2O_3$, $K_2O$ and $Na_2O$, CaO and MgO contents in red beds, igneous rocks, and metamorphic rocks, respectively. Figure 5 shows the comparison of $SiO_2$ and $Al_2O_3$, FeO and $Fe_2O_3$, $K_2O$ and $Na_2O$, CaO and MgO contents in red beds and other sedimentary rocks respectively.

The content of $SiO_2$ in the red beds is about 30%~80%, $Al_2O_3$ is about 8%~30%, $Fe_2O_3$ is about



0%~10%, FeO is about 0%~3%, $K_2O$ is about 0%~10%, $Na_2O$ is about 0%~2.5%, CaO is about
0%~10%, and MgO is about 0%~5%. Compared with igneous rocks, metamorphic rocks, and other
sedimentary rocks, the content of each chemical composition of the red beds has three relationships
with the content of corresponding chemical composition of other rocks: inclusion relationship (the data
distribution range of one rock completely covers and is larger than the data range of the other rock),
intersection relationship (the data distribution range of one rock intersects with the data distribution
range of another rock), and mutual difference relationship (the data distribution range of one rock does
not intersect at all with the data distribution range of another rock). The distribution range of $SiO_2$ and
$Al_2O_3$ content in the red beds includes the distribution range of $SiO_2$ and $Al_2O_3$ content in 9 types of
rocks, namely andesite, basalt, diorite, granite, black shale, claystone, mudstone, siliciclastic, and tuff.
The distribution range of $SiO_2$ and $Al_2O_3$ content in the red beds intersects with that in breccia, lignite,
and marl. The distribution range of $SiO_2$ and $Al_2O_3$ content in gneiss, marble, arkose, dolomite, and
limestone is different from that in the red beds. The distribution range of $Fe_2O_3$ and FeO content in the
red beds includes the distribution range of $Fe_2O_3$ and FeO content in granite, marble, and lignite. The
distribution range of $Fe_2O_3$ and FeO content in the red beds intersects with that in 8 kinds of rocks,
namely, andesite, basalt, diorite, breccia, claystone, dolomite, limestone, and mudstone. The
distribution range of $Fe_2O_3$ and FeO content in gneiss, arkose, black shale, siliciclastic, and tuff is
different from that in the red beds. The distribution range of $K_2O$ and $Na_2O$ content in the red beds
includes the distribution range of $K_2O$ and $Na_2O$ content in lignite. The distribution range of $K_2O$ and
$Na_2O$ content in the red beds intersects with that in 15 kinds of rocks, including andesite, basalt, diorite,
granite, marble, arkose, black shale, breccia, claystone, dolomite, limestone, marl, mudstone,
siliciclastic, and tuff. The distribution range of $K_2O$ and $Na_2O$ content in gneiss is different from that in
the red beds. The distribution range of CaO and MgO content in the red beds includes the distribution
range of CaO and MgO content in granite, black shale, and lignite. The distribution range of CaO and
MgO content in the red beds intersects with that in 13 types of rocks, including andesite, basalt, diorite,
gneiss, arkose, breccia, claystone, dolomite, limestone, marl, mudstone, siliciclastic, and tuff. The
distribution range of CaO and MgO content in marble is different from that in the red beds. Therefore,
from a qualitative perspective, it can be seen that the red beds differ in chemical composition from 8



kinds of rocks, including gneiss, marble, arkose, dolomite, limestone, black-shale, siliciclastic, and tuff,
and also intersects with other rocks to varying degrees. But this is not enough as a criterion to determine
the difference between red beds and other rocks.

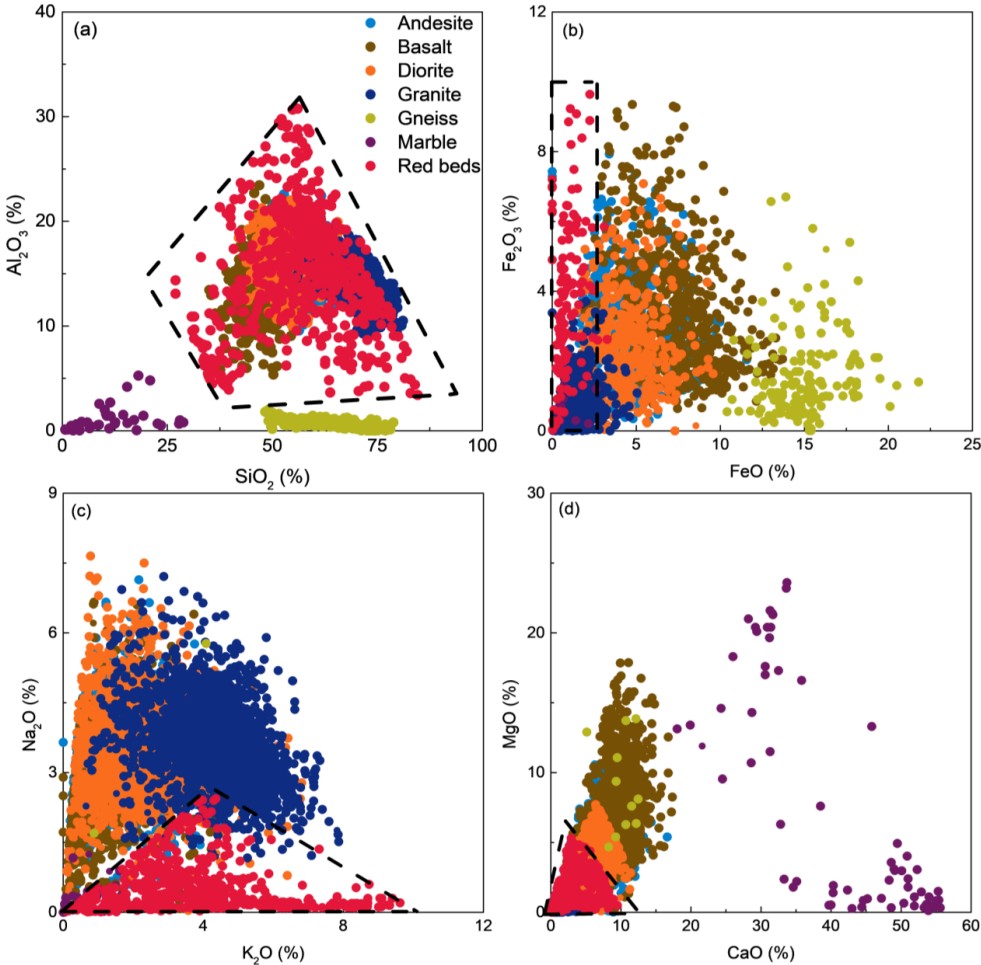


**Figure 4.** Comparison of (a) $SiO_2$ and $Al_2O_3$, (b) FeO and $Fe_2O_3$, (c) $K_2O$ and $Na_2O$, (d) CaO and

MgO contents in red beds, igneous rock, and metamorphic rocks, respectively.

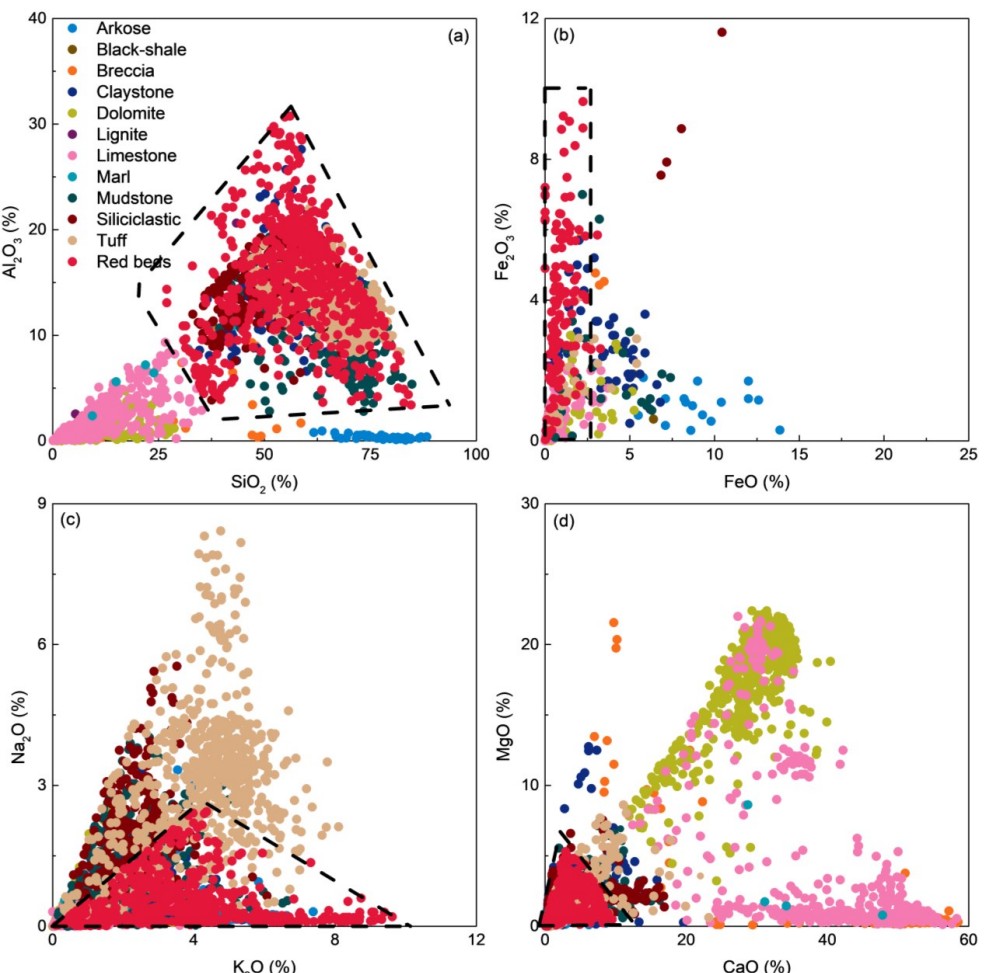

**Figure 5.** Comparison of (a) $SiO_2$ and $Al_2O_3$, (b) FeO and $Fe_2O_3$, (c) $K_2O$ and $Na_2O$, (d) CaO and MgO contents in red beds and other sedimentary rocks respectively.

Figures 6~7 mainly analyze the differences in chemical compositions between red beds and other rocks through further data statistics and box plots of the scatter plots mentioned above, and propose quantitative identification criterion for the red beds chemical compositions combination. The red dashed box in the figure represents rocks that differ from the red beds data, while the black dashed box represents rocks that intersect less than 25% with the red beds data. The data collected in section 2.1 comes from published papers or databases, and its accuracy and robustness have been explained in



relevant literature. In order to ensure the exclusion of outliers in the box plots mentioned above during
the analysis of this study. The horizontal gray dashes corresponding to the red beds box chart represent
10% percentile (the same below), lower quartile (25% percentile), median (50% percentile), upper
quartile (75% percentile), and 90% percentile in the red beds data from bottom to top. Figure 6 shows
the chemical compositions combination comparison of $SiO_2+Al_2O_3$ (total content, the same below) and
$Al_2O_3/SiO_2$ (content ratio, the same below), $FeO+Fe_2O_3$ and $Fe_2O_3/FeO$, $K_2O+Na_2O$ and $Na_2O/K_2O$,
$CaO+MgO$ and $MgO/CaO$ in red beds, igneous rock, and metamorphic rocks, respectively. Figure 7
respectively shows the chemical compositions combination comparison of $SiO_2+Al_2O_3$ and $Al_2O_3/SiO_2$,
$FeO+Fe_2O_3$ and $Fe_2O_3/FeO$, $K_2O+Na_2O$ and $Na_2O/K_2O$, $CaO+MgO$ and $MgO/CaO$ in red beds and
other sedimentary rocks.

The $SiO_2+Al_2O_3$ content in the red beds is 54.7%~85.0% (10%~90% percentile, the same below),

the $Al_2O_3/SiO_2$ ratio is 0.14~0.41, the $FeO+Fe_2O_3$ content is 0.9%~7.9%, the $Fe_2O_3/FeO$ ratio is
1.52~7.70, the $K_2O+Na_2O$ content is 1.6%~6.8%, the $Na_2O/K_2O$ ratio is 0.02~0.43, the $CaO+MgO$
content is 0.8%~9.2%, and the $MgO/CaO$ ratio is 0.16~1.57. By comparing the content of $SiO_2+Al_2O_3$,
the red beds are distinct or have small intersections (less than 25%, the same below) with granite, marble,
dolomite, lignite, limestone, and marl. By comparing the $Al_2O_3/SiO_2$ ratio, it is found that the red beds
are distinct or have small intersections with gneiss, marble, arkose, and lignite. By comparing the
content of $FeO+Fe_2O_3$, it is found that the red beds are distinct or have small intersections with basalt,
gneiss, arkose, and siliciclastic. By comparing the $Fe_2O_3/FeO$ ratio, it is found that the red beds are
distinct or have small intersections with andesite, basalt, diorite, granite, gneiss, marble, arkose, black
shale, dolomite, mudstone, siliclastic, and tuff. Through the comparison of $K_2O+Na_2O$ content, the red
beds are distinct or have small intersections with granite, marble, breccia, dolomite, and limestone. By
comparing the $Na_2O/K_2O$ ratio, the red beds are distinct or have small intersections with andesite, basalt,
diorite, gneiss, lignite, siliciclastic, and tuff. Through the comparison of $CaO+MgO$ content, the red
beds are distinct or have small intersections with andesite, basalt, gneiss, marble, breccia, dolomite,
limestone, and marl. By comparing the $MgO/CaO$ ratio, it is difficult to distinguish the red beds from
other rocks.





**Figure 6.** Chemical compositions comparison of (a) $SiO_2+Al_2O_3$, $Al_2O_3/SiO_2$, (b) $FeO+Fe_2O_3$, $Fe_2O_3/FeO$,

(c) $K_2O+Na_2O$, $Na_2O/K_2O$, (d) $CaO+MgO$, $MgO/CaO$ in red beds, igneous rock, and metamorphic rocks.



**Figure 7.** Chemical compositions comparison of (a) $SiO_2+Al_2O_3$, $Al_2O_3/SiO_2$, (b) $FeO+Fe_2O_3$, $Fe_2O_3/FeO$,

(c) $K_2O+Na_2O$, $Na_2O/K_2O$, (d) $CaO+MgO$, $MgO/CaO$ in red beds and other sedimentary rocks.






In summary, there are differences in chemical compositions between red beds and other rocks, and
the use of chemical compositions combination rules can serve as a quantitative criterion for identifying
red beds. Simultaneously meeting the following chemical compositions combinations as a quantitative
criterion to distinguish red beds with different geological ages and various lithologies from other rocks:
$SiO_2+Al_2O_3$ ≈ 50.7%~85.0%, $Al_2O_3/SiO_2$ ≈ 0.14~0.41, $FeO+Fe_2O_3$ ≈ 0.9%~7.9%,
$Fe_2O_3/FeO$ ≈ 1.52~7.70, $K_2O+Na_2O$ ≈ 1.6%~6.8%, $Na_2O/K_2O$ ≈ 0.02~0.43,
$CaO+MgO$ ≈ 0.8%~9.2%.

3.4 Red beds identification quantization criterion verification
The chemical composition combinations of the 15 selected rocks in this study are shown in Table
3. The chemical composition combinations of 12 kinds of red beds are all within the scope of the
quantitative criterion. There are some chemical composition combinations of the 3 non-red beds
sedimentary rocks that are outside the scope of the red beds quantitative criterion (the numbers in bold
and underlined in the table). For example, the content of $SiO_2+Al_2O_3$, $FeO+Fe_2O_3$, $K_2O+Na_2O$ in
limestone is lower than the range of quantification criterion, while the content of $CaO+MgO$ in
limestone is higher than the range of quantification criterion; $Fe_2O_3/FeO$ and $K_2O+Na_2O$ in arkose are
below the quantification criterion; $SiO_2+Al_2O_3$ and $Na_2O/K_2O$ in mudstone are higher than the
quantification criterion, while $Fe_2O_3/FeO$ and $K_2O+Na_2O$ are lower than the quantification criterion.
This is consistent with the research results in Figure 7, once again proving the reliability of the
quantification criterion proposed in this study.
**Table 3.** Chemical composition combinations of 15 kinds of rocks.

| No. | $SiO_2$ (%) | $Al_2O_3$ (%) | $Fe_2O_3$ (%) | FeO (%) | $Na_2O$ (%) | $K_2O$ (%) | MgO (%) | CaO (%) | $SiO_2+$ $Al_2O_3$ (%) | $Al_2O_3/$ $SiO_2$ | $FeO+$ $Fe_2O_3$ (%) | $Fe_2O_3/$ FeO | $K_2O+$ $Na_2O$ (%) | $Na_2O/$ $K_2O$ | $CaO+$ $MgO$ (%) |
|---|---|---|---|---|---|---|---|---|---|---|---|---|---|---|---|
| 1 | 43.3 | 15.0 | 2.9 | 0.7 | 0.2 | 1.9 | 3.3 | 1.1 | 58.3 | 0.35 | 3.6 | 4.12 | 2.1 | 0.10 | 4.4 |
| 2 | 45.8 | 18.3 | 4.1 | 1.0 | 0.3 | 2.6 | 2.3 | 0.0 | 64.1 | 0.40 | 5.1 | 4.12 | 2.9 | 0.10 | 2.3 |
| 3 | 40.1 | 15.5 | 3.7 | 0.9 | 0.2 | 2.1 | 3.6 | 0.0 | 55.6 | 0.39 | 4.6 | 4.12 | 2.3 | 0.10 | 3.6 |
| 4 | 48.8 | 14.3 | 3.1 | 0.7 | 0.3 | 2.9 | 2.9 | 6.1 | 63.1 | 0.29 | 3.8 | 4.12 | 3.2 | 0.10 | 9.0 |



| 5 | 62.0 | 15.8 | 2.7 | 0.6 | 0.3 | 3.2 | 3.1 | 0.0 | 77.8 | 0.26 | 3.3 | 4.12 | 3.5 | 0.10 | 3.1 |
| 6 | 42.8 | 9.4 | 1.6 | 0.4 | 0.2 | 1.5 | 0.4 | 4.1 | 52.2 | 0.22 | 2.0 | 4.12 | 1.7 | 0.10 | 4.5 |
| 7 | 52.2 | 17.1 | 1.5 | 0.4 | 0.2 | 2.3 | 2.5 | 0.0 | 69.3 | 0.33 | 1.9 | 4.12 | 2.5 | 0.10 | 2.5 |
| 8 | 58.3 | 18.6 | 1.6 | 0.4 | 0.2 | 1.9 | 4.0 | 0.8 | 76.9 | 0.32 | 2.0 | 4.12 | 2.1 | 0.10 | 4.8 |
| 9 | 39.9 | 11.2 | 1.3 | 0.3 | 0.2 | 1.5 | 3.9 | 0.0 | 51.1 | 0.28 | 1.4 | 4.12 | 1.7 | 0.10 | 3.9 |
| 10 | 48.2 | 9.6 | 1.0 | 0.2 | 0.2 | 2.4 | 3.5 | 1.9 | 57.8 | 0.20 | 1.2 | 4.12 | 2.6 | 0.10 | 5.4 |
| 11 | 50.5 | 14.2 | 2.1 | 0.5 | 0.2 | 2.3 | 0.8 | 5.1 | 64.7 | 0.28 | 2.6 | 4.12 | 2.5 | 0.10 | 5.9 |
| 12 | 45.1 | 8.4 | 3.5 | 0.8 | 0.2 | 2.0 | 2.3 | 1.6 | 53.5 | 0.19 | 4.3 | 4.12 | 2.2 | 0.10 | 3.9 |
| 13 | 13.6 | 2.3 | 0.1 | 0.1 | 0.2 | 0.5 | 3.2 | 39.6 | *15.9* | 0.17 | *0.2* | 1.23 | *0.7* | 0.33 | *42.8* |
| 14 | 56.9 | 14.9 | 0.3 | 2.3 | 0.2 | 1.3 | 3.3 | 1.1 | 71.8 | 0.26 | 2.6 | 0.11 | *1.5* | 0.18 | 4.4 |
| 15 | 69.7 | 21.2 | 0.6 | 0.7 | 0.3 | 0.5 | 0.9 | 0.0 | *90.9* | 0.30 | 1.3 | 0.87 | *0.8* | 0.61 | 0.9 |


3.5 Research results application methods

Figure 8 shows the application methods of the research results. According to the methods for

emergency management of landslide geological disasters (Fu et al. 2021), landslide risk assessment
(including risk identification, risk analysis, and risk assessment) and risk management (developing and
selecting treatment plans, as well as planning, implementing, and evaluating treatment methods) need
to be carried out before the landslide occurs. In the field of engineering geology, risk identification is
the most important prerequisite for landslide emergency response. Red beds is the slippery layer that
needs to be identified in risk identification.

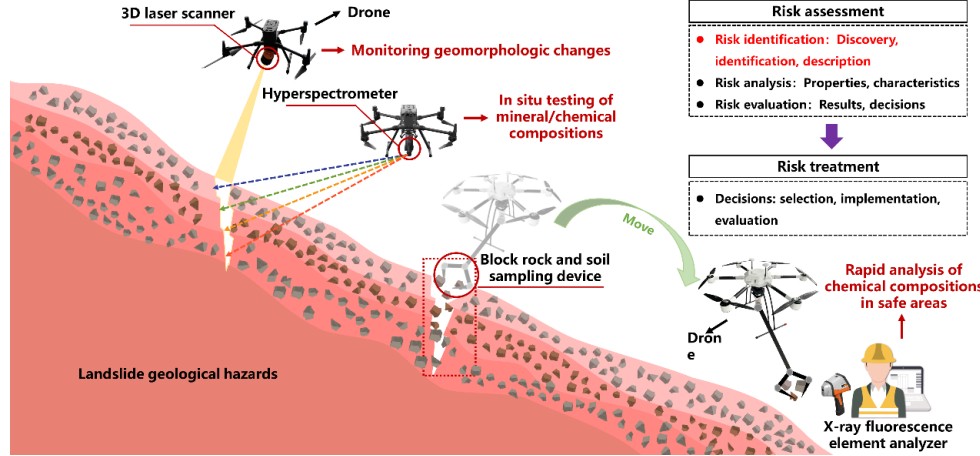


**Figure 8.** Research results used for risk identification.




At present, the commonly used risk identification method is to use drones to carry image capture
devices for three-dimensional reconstruction of slope images, determine the volume of landslide
accumulation, and determine the shape changes of the slope (Chen et al. 2020; Fu et al. 2021), which
can be also used for mountain rescue (Wankmuller et al. 2021). Based on the drone technology,
combined with the Optech Polaris LR 3D laser scanner and the HY-9070 hyperspectral analyzer of Sun
Yat-sen University, the landslide shape change and remote monitoring of mineral and chemical
compositions can be realized to identify whether it is a red beds landslide. It can also use a drone
equipped with a block rock and soil sampling device to collect representative blocks of rock and soil
within cracks to a safe area, and then use the MiX5 Pro handheld X-ray fluorescence element analyzer
for rapid analysis. Therefore, the research results can be used for rapid identification of red beds,
achieving risk assessment and rapid response of geological disasters such as landslides.

**4. Conclusions**
(1) In response to the rapid identification of red beds in geological disaster emergency response, a
rapid quantitative identification criterion based on the basic chemical compositions combination rules
of red beds has been established, taking into account the correlation between red beds geomorphic
characteristics, mineral compositions, and chemical compositions.
(2) The results indicate that the red beds in the geomorphic characteristics has obvious interlayer
characteristics and its appearance is red. In mineral composition, the ratio of clay minerals to other
minerals of red beds ranges from 0.11 to 1.50, and the content of hematite of red beds ranges from 1.5%
to 10.0%. The following chemical composition combinations can be used as red beds quantification
criterion: $SiO_2+Al_2O_3 \approx 50.7\%\sim85.0\%$, $Al_2O_3/SiO_2 \approx 0.14\sim0.41$, $FeO+Fe_2O_3 \approx 0.9\%\sim7.9\%$,
$Fe_2O_3/FeO \approx 1.52\sim7.70$, $K_2O+Na_2O \approx 1.6\%\sim6.8\%$, $Na_2O/K_2O \approx 0.02\sim0.43$,
$CaO+MgO \approx 0.8\%\sim9.2\%$. The reliability of the quantitative criterion proposed by this study was
verified by collecting 15 kinds of rocks and analyzing their chemical composition combinations.
(3) The combination of research results with existing landslide geological hazard risk identification
techniques can effectively carry out rapid response to geological disasters, which is very important for



emergency response to geological disasters. Moreover, the research results can also be applied to the
quantitative identification of red beds in other fields such as resources, ecology, environment, energy,
materials, etc.

**Declarations**
**Availability of data and materials**
The data that support the findings of this study are available in supplementary materials.
**Competing interests**
The authors declare no conflict of interest. The funders had no role in the design of the study; in
the collection, analyses, or interpretation of data; in the writing of the manuscript, or in the decision to
publish the results.
**Funding**
The research is supported by the National Natural Science Foundation of China (NSFC) (Grant
Numbers: 42293354, 42293351, 42293355, 42277131, 41977230).
**Authors' contributions**
Conceptualization, C.Z. and Z.L.; methodology, G.C. and Z.L.; software, G.C. and L.K.; validation,
G.C., L.K., and Z.L.; formal analysis, C.Z. and Z.L.; investigation, G.C., J.L., and L.Y.; resources, G.C.
and L.K.; data curation, G.C., J.L., L.Y. and L.K.; writing—original draft preparation, G.C. and L.K.;
writing—review and editing, G.C., Z.L., and L.Z.; visualization, L.Y.; supervision, Z.L. and L.Z.;
project administration, C.Z.; funding acquisition, C.Z. All authors have read and agreed to the published
version of the manuscript.
**Acknowledgments**
The authors would like to thank the anonymous reviewers for their very constructive and helpful
comments.
**Supplementary Materials**
Supplementary Table 1: Mineral compositions of the red beds.
Supplementary Table 2: Chemical compositions of the red beds.
Supplementary Table 3: Chemical compositions of the andesite.



Supplementary Table 4: Chemical compositions of the basalt.

Supplementary Table 5: Chemical compositions of the diorite.

Supplementary Table 6: Chemical compositions of the granite.

Supplementary Table 7: Chemical compositions of the gneiss.

Supplementary Table 8: Chemical compositions of the marble.

Supplementary Table 9: Chemical compositions of the arkose.

Supplementary Table 10: Chemical compositions of the black-shale.

Supplementary Table 11: Chemical compositions of the breccia.

Supplementary Table 12: Chemical compositions of the claystone.

Supplementary Table 13: Chemical compositions of the dolomite.

Supplementary Table 14: Chemical compositions of the lignite.

Supplementary Table 15: Chemical compositions of the limestone.

Supplementary Table 16: Chemical compositions of the marl.

Supplementary Table 17: Chemical compositions of the mudstone.

Supplementary Table 18: Chemical compositions of the siliciclastic.

Supplementary Table 19: Chemical compositions of the tuff.

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
