# Peer review of "Basic chemical compositions combination rules and quantitative criterion of red beds"

_EGUsphere, 2023_

## Referee Comment (RC1)

**Comments and Suggestions for Authors**

This study established a rapid quantitative identification criterion based on the basic chemical compositions combination rules in the red beds. The manuscript is well written and the topic is interesting and novel. By comparing the chemical composition combinations of 15 kinds of rocks collected from China in this study, it is proven that the quantitative criterion proposed in this study are effective. The work performed in the manuscript is comprehensive and in-depth. However, there are some minor problems that should be addressed before any further publication process.

**Required changes:**

**Point 1:** In fact, the second section on methods is very rich in content. It mentioned previous research on the comparison between red beds and other rocks. But in this chapter, whether this part of the content overlaps with the introduction. Please provide a reasonable explanation. Additionally, as this section is too lengthy, it is recommended to add a suitable flowchart to facilitate better reading for readers. In addition, in section 2.1, the principle and instrument photos of the handheld elemental analyzer appear slightly monotonous. A schematic diagram can be formed by combining its principle and instrument photos.

**Point 2:** In the third section of Results and Discussion, both sections 3.2 and 3.3 describe the results through tables or figures. In section 3.1, it was found that the geomorphic characteristics of the red beds were mainly described through literature review, and the content was not rich enough. Can appropriate figures or table data be added.

**Point 3:** Some general recommendations regarding the presentation of contents. (1) figure 2. I suggest writing "Location of the study area" and, if possible, add a description of what there is in the photo. For example, can the sample be dispersed by adding a map of China. (2) In the research status of the introduction, it should take a positive attitude towards the previous research results on the whole and deny them praise. It could talk about advantages before disadvantages. (3) Do the colors in the small figures

in Figures 4 and 5 represent the same type of rock? Please make a note or add a picture frame in the image. (4) Can Table 3 also be presented in the form of an interval graph, and then 15 types of rocks can be scattered in this interval for intuitive judgment and verification.

**Point 4:** Problems in the tables and figures in the manuscript. Please carefully check the accuracy of the units and formats of all tables and figures. Such as, the format of each tables and figures needs to be consistent before and after.

**Point 5:** Some general recommendations regarding the contents of the paper. I found that the authors do not use the unified and correct significant figures and decimals in the whole paper (e.g., 43.5 and 0.40 in the table 3; 2.0% and 30% in the figure 6). It is necessary to correct the whole paper for accuracy and precision. And item 1 in conclusion is too general and appears to be not connected well to the presented results. Has the current problem been resolved, please either remove or be more specific. Minor grammatical problems in the full text need to be corrected (e.g., an adequate-fitting prediction method in the abstract).

---

## Author Response (AR1)

**Reviewer 1:**

Comments and Suggestions for Authors

This study established a rapid quantitative identification criterion based on the basic chemical compositions combination rules in the red beds. The manuscript is well written and the topic is interesting and novel. By comparing the chemical composition combinations of 15 kinds of rocks collected from China in this study, it is proven that the quantitative criterion proposed in this study are effective. The work performed in the manuscript is comprehensive and in-depth. However, there are some minor problems that should be addressed before any further publication process.

**Response:** Thank you very much for your review of this manuscript. The following are the revisions made to the manuscript one by one based on your comments.

Required changes:

**Point 1:** In fact, the second section on methods is very rich in content. It mentioned previous research on the comparison between red beds and other rocks. But in this chapter, whether this part of the content overlaps with the introduction. Please provide a reasonable explanation. Additionally, as this section is too lengthy, it is recommended to add a suitable flowchart to facilitate better reading for readers. In addition, in section 2.1, the principle and instrument photos of the handheld elemental analyzer appear slightly monotonous. A schematic diagram can be formed by combining its principle and instrument photos.

**Response 1:** We have revised the last paragraph of **1 Introduction** based on your suggestion and adjusted the content of the paragraph regarding to **2 Methods**. For the convenience of readers to read better, adjust the position of Figure 1 to **2 Methods** according to your suggestions. The schematic diagram of the handheld elemental analyzer has been added (Figure 3).

**1. Introduction          Lines 67-70**

*"Therefore, the purpose of this study to develop a quantitative criterion for quickly and accurately identifying the red beds. This study first collected the data about the geomorphic characteristics, mineral content, and chemical composition of red beds and other rocks, then compared these data to obtain the basic characteristics of red beds, and finally summarized and analyzed the red beds identification criterion and verified the reliability of this criterion."*

**2. Methods      Lines 73-77**

*"Figure 1 shows the methodology used in this study involving the investigation of geomorphic characteristics, mineral compositions, and chemical compositions (the perspective of chemical compositions is the focus of this study). In this study, data on geomorphological features, mineral content and chemical composition of the red beds and other rocks were first collected, then these data were compared to derive the basic characteristics of the red beds, and finally the red bed identification criteria were summarized and analyzed, and the reliability of the criteria was verified."*

**2.2 Criterion verification      Lines 152-156**

*"The working principle of this instrument is that a miniature X-ray source provides tube voltage and tube current, and the light tube emits continuous X-ray spectral lines. The X-rays irradiated on the sample knock out the inner electrons of the K and L layers of the element atoms, and the holes in the low-energy layer are filled by high-energy outer electrons (N layer). The high-energy electrons emit excess energy as X-ray fluorescence (Kα) with elemental characteristics. Thus, the instrument detects the type and concentration of elements through the emitted spectral lines."*

[Figure]

**Figure 3: MiX5 Pro handheld X-ray fluorescence element analyzer.**

**Point 2:** In the third section of Results and Discussion, both sections 3.2 and 3.3 describe the results through tables or figures. In section 3.1, it was found that the geomorphic characteristics of the red beds were mainly described through literature review, and the content was not rich enough. Can appropriate figures or table data be added.

**Response 2:** In Section 3.1, the identification of red beds is qualitatively explained by adding Figure 4 to illustrate the geomorphic characteristics of the red beds.

[Figure]

[Figure]

Red beds Yadan landform          Red beds Danxia landform

**Figure 4: Geomorphic characteristics of the red beds.**

**Point 3:** Some general recommendations regarding the presentation of contents. (1) figure 2. I suggest writing "Location of the study area" and, if possible, add a description of what there is in the photo. For example, can the sample be dispersed by adding a map of China. (2) In the research status of the introduction, it should take a positive attitude towards the previous research results on the whole and deny them praise. It could talk about advantages before disadvantages. (3) Do the colors in the small figures in Figures 4 and 5 represent the same type of rock? Please make a note or add a picture frame in the image. (4) Can Table 3 also be presented in the form of an interval graph, and then 15 types of rocks can be scattered in this interval for intuitive judgment and verification.

**Response 3:** (1) Sample locations have been added to Figure 2 and labeled on the map of China. (2) The Introduction provides a positive evaluation of previous achievements and points out the research that still needs to be carried out in this manuscript. (3) The same legend in Figures 4 and 5 represents the same type of rock, as explained in the figure title. (4) Table 3 has been modified to indicate the differences between these rocks and the red layer through orange shading.

[Figure]

**Figure 2: Distribution areas of red beds in China and sampling locations for 15 types of rocks.**

**1 Introduction    Lines 55-56**

*"At this stage, the research on the geomorphology, mineral color and clay content of the red beds lays the foundation for the identification of the red beds, but this identification is still vague and needs to be further quantified."*

**Figure 6: Comparison of (a) SiO₂ and Al₂O₃, (b) FeO and Fe₂O₃, (c) K₂O and Na₂O, (d) CaO and MgO contents in red beds, igneous rock, and metamorphic rocks, respectively (Note: Icons of the same color in the figure have the same meanings).**

**Table 3: Chemical composition combinations of 15 kinds of rocks.**

| No. | SiO$_2$ (%) | Al$_2$O$_3$ (%) | Fe$_2$O$_3$ (%) | FeO (%) | Na$_2$O (%) | K$_2$O (%) | MgO (%) | CaO (%) | SiO$_2$+ Al$_2$O$_3$ (%) | Al$_2$O$_3$/ SiO$_2$ | FeO+ Fe$_2$O$_3$ (%) | Fe$_2$O$_3$/ FeO | K$_2$O+ Na$_2$O (%) | Na$_2$O/ K$_2$O | CaO+ MgO (%) |
|---|---|---|---|---|---|---|---|---|---|---|---|---|---|---|---|
| 1 | 43.3 | 15.0 | 2.9 | 0.7 | 0.2 | 1.9 | 3.3 | 1.1 | 58.3 | 0.35 | 3.6 | 4.12 | 2.1 | 0.10 | 4.4 |
| 2 | 45.8 | 18.3 | 4.1 | 1.0 | 0.3 | 2.6 | 2.3 | 0.0 | 64.1 | 0.40 | 5.1 | 4.12 | 2.9 | 0.10 | 2.3 |
| 3 | 40.1 | 15.5 | 3.7 | 0.9 | 0.2 | 2.1 | 3.6 | 0.0 | 55.6 | 0.39 | 4.6 | 4.12 | 2.3 | 0.10 | 3.6 |
| 4 | 48.8 | 14.3 | 3.1 | 0.7 | 0.3 | 2.9 | 2.9 | 6.1 | 63.1 | 0.29 | 3.8 | 4.12 | 3.2 | 0.10 | 9.0 |
| 5 | 62.0 | 15.8 | 2.7 | 0.6 | 0.3 | 3.2 | 3.1 | 0.0 | 77.8 | 0.26 | 3.3 | 4.12 | 3.5 | 0.10 | 3.1 |
| 6 | 42.8 | 9.4 | 1.6 | 0.4 | 0.2 | 1.5 | 0.4 | 4.1 | 52.2 | 0.22 | 2.0 | 4.12 | 1.7 | 0.10 | 4.5 |
| 7 | 52.2 | 17.1 | 1.5 | 0.4 | 0.2 | 2.3 | 2.5 | 0.0 | 69.3 | 0.33 | 1.9 | 4.12 | 2.5 | 0.10 | 2.5 |
| 8 | 58.3 | 18.6 | 1.6 | 0.4 | 0.2 | 1.9 | 4.0 | 0.8 | 76.9 | 0.32 | 2.0 | 4.12 | 2.1 | 0.10 | 4.8 |
| 9 | 39.9 | 11.2 | 1.3 | 0.3 | 0.2 | 1.5 | 3.9 | 0.0 | 51.1 | 0.28 | 1.4 | 4.12 | 1.7 | 0.10 | 3.9 |
| 10 | 48.2 | 9.6 | 1.0 | 0.2 | 0.2 | 2.4 | 3.5 | 1.9 | 57.8 | 0.20 | 1.2 | 4.12 | 2.6 | 0.10 | 5.4 |
| 11 | 50.5 | 14.2 | 2.1 | 0.5 | 0.2 | 2.3 | 0.8 | 5.1 | 64.7 | 0.28 | 2.6 | 4.12 | 2.5 | 0.10 | 5.9 |
| 12 | 45.1 | 8.4 | 3.5 | 0.8 | 0.2 | 2.0 | 2.3 | 1.6 | 53.5 | 0.19 | 4.3 | 4.12 | 2.2 | 0.10 | 3.9 |
| 13 | 13.6 | 2.3 | 0.1 | 0.1 | 0.2 | 0.5 | 3.2 | 39.6 | **15.9** | 0.17 | **0.2** | 1.23 | **0.7** | 0.33 | **42.8** |
| 14 | 56.9 | 14.9 | 0.3 | 2.3 | 0.2 | 1.3 | 3.3 | 1.1 | 71.8 | 0.26 | 2.6 | 0.11 | **1.5** | 0.18 | 4.4 |
| 15 | 69.7 | 21.2 | 0.6 | 0.7 | 0.3 | 0.5 | 0.9 | 0.0 | **90.9** | 0.30 | 1.3 | 0.87 | **0.8** | 0.61 | 0.9 |

**Point 4:** Problems in the tables and figures in the manuscript. Please carefully check the accuracy of the units and formats of all tables and figures. Such as, the format of each tables and figures needs to be consistent before and after.

**Response 4:** We have standardized the format of all figures and tables in the manuscript based on your suggestions.

**Point 5:** Some general recommendations regarding the contents of the paper. I found that the authors do not use the unified and correct significant figures and decimals in the whole paper (e.g., 43.5 and 0.40 in the Table 3; 2.0% and 30% in the Figure 6). It is necessary to correct the whole paper for accuracy and precision. And item 1 in conclusion is too general and appears to be not connected well to the presented results. Has the current problem been resolved, please either remove or be more specific. Minor grammatical problems in the full text need to be corrected.

**Response 5:** (1) We have reviewed and revised all the figures and tables in the manuscript. The component content and its sum in Table 3 are expressed to one decimal place, while the component content ratio is expressed to two decimal places; The sum of component content in Figure 6 is expressed as a percentage, while the ratio of component content is expressed as a numerical value. (2) The first conclusion item in the manuscript has been supplemented. (3) The grammar of the manuscript has been

checked and revised.

**4. Conclusions      Line 325**

*"(1)…… It solves the current problem of fuzzy identification of the red beds."*

**Reviewer 1:**

Cui et al.'s research aimed to address the issues of slow recognition speed and inaccurate recognition results of red beds. Based on the correlation between red beds geomorphic characteristics, mineral compositions, and chemical compositions, a rapid quantitative identification standard was established based on the basic chemical composition combination law of red beds. The research content of this manuscript is very interesting, the research methods are reasonable, and the research results are very valuable. However, there are still some minor issues with the manuscript. The following modifications are suggested for the manuscript to be published in this journal:

**Response:** Thank you very much for the reviewer's review. We have made revisions to the following issues item by item based on the feedback provided.

-Lines 28-30: After verifying the quantification criterion for red beds recognition, it is necessary to further explain the application scope or prospects of this criterion.

**Response 1:** The application scope and prospects of the quantification criterion for red beds recognition have been added in the last sentence of the abstract.

**Abstract**

*"The study results can be used for rapid identification of red beds, achieving risk assessment and rapid response of geological disasters such as landslides."*

-Lines 76-77: The results of Moonjun et al. cited should be placed in Lines 51-52 to illustrate the relationship between geomorphic characteristics, mineral compositions, and chemical compositions.

**Response 2:** This sentence has been revised and placed in the second paragraph of the Introduction.

**1. Introduction**

*"And, there is a close relationship between these perspectives (Moonjun et al., 2017; Bankole et al., 2016; Perri et al., 2013)."*

-Lines 88-90: The description of this sentence is not accurate. The data of red beds and other rocks not only comes from previous studies, but also from some database. It is recommended to modify or delete it.

**Response 3:** This sentence has been deleted from the manuscript.

-Lines 100-101: This statement regarding the relationship between geomorphic characteristics and mineral compositions should be explained before Lines 51-52.

**Response 4:** This sentence has been revised and placed in the second paragraph of the Introduction.

**1. Introduction**

*"At present, the studies on red beds identification are mostly carried out from the perspectives of geomorphic characteristics, mineral compositions, and chemical compositions (Cui et al., 2022; Zhou et al., 2021). And, there is a close relationship between these perspectives (Moonjun et al., 2017; Bankole et al., 2016; Perri et al., 2013)."*

-Lines 111-112: This statement regarding the relationship between chemical compositions and mineral compositions should be explained before Lines 51-52.

**Response 5:** This sentence has been revised and placed in the second paragraph of the Introduction.

**1. Introduction**

*"At present, the studies on red beds identification are mostly carried out from the perspectives of geomorphic characteristics, mineral compositions, and chemical compositions (Cui et al., 2022; Zhou et al., 2021). And, there is a close relationship between these perspectives (Moonjun et al., 2017; Bankole et al., 2016; Perri et al., 2013). For example, the content of $Fe_2O_3$ or hematite in the red beds is higher than that in the grey beds (Hu et al., 2006)."*

-Line 122: The "." in parentheses should be deleted, please check the entire text.

**Response 6:** All the "." in this paragraph have been removed.

-Lines 164-168: After sampling, please explain the processing flow of the sample before conducting the test.

**Response 7:** The process of conducting on-site testing on samples after sampling has been explained in the manuscript.

**2.2 Criterion verification**

*"After on-site sampling, use a hammer to smash the rock block out of the fresh surface. Then, the fresh surface was analyzed using the MiX5 Pro handheld X-ray fluorescence element analyzer (Figure 3) from*

*Sun Yat-sen University to check whether these elements conform to the basic chemical compositions combination rules of red beds proposed by this study."*

-Lines 201-202: It is necessary to clarify the specific meaning of "geomorphic characteristics" here.

**Response 8:** The specific meaning of "geometric characteristics" has been explained in detail in the manuscript.

3.1 Geomorphic characteristics of red beds

*"Compared with the obvious layering and red appearance characteristics of red beds, igneous rocks and metamorphic rocks do not show the two characteristics of red appearance and bedding at the same time."*

-Lines 331-332: Here, it is necessary to provide a detailed explanation of what rocks "non red beds sedimentary rocks" refer to.

**Response 9:** The specific types of "non red beds sedimentary rocks" have been explained in detail in the manuscript.

**3.4 Red beds identification quantization criterion verification**

*"There are some chemical composition combinations of the 3 non-red beds sedimentary rocks (limestone, arkose, and mudstone) that are outside the scope of the red beds quantitative criterion (the numbers in bold and underlined in the table)."*

-Line 376: "by" should be changed to "in", please check the grammar of the entire text.

**Response 10:** The grammar of the manuscript has been checked and revised.

---

## Author Response (AR2)

The authors have performed an extensive review of literature data on sedimentary "red beds" in order to highlight their mineralogical and chemical signature, with respect to global databases on igneous, metamorphic and sedimentary rocks. The review shows that the "red beds" span a huge range in mineralogical and chemical composition. In terms of major elements, the most distinctive feature is the high Fe2O3/FeO ratio and the low Na2O/K2O ratio; However, as correctly acknowledged by the authors, XRF (hand held or lab) does not permit to quantify the iron oxidation state and this technique has very poor precision on light elements like Na2O. This make the application of the in situ analysis very tricky/impossible, as the main discriminant factors cannot be quantified by XRF; The analysis is very preliminary and based on basic statistics of rock composition in terms of major elements. More complete statistical analysis (for instance, PCA) could permit better to quantify the combination of major elements potentially specific of these rocks, in spite of their highly variable signature. Moreover, trace elements or other chemical signatures (stable isotopes) might result to be more reliable to identify red beds, for instance; In spite of the above-mentioned critical limitations of the current study, the two reviews have resulted in surprisingly minor modifications of the initial submission and cannot be considered as conclusive. I suggest paper re-submission after substantial modification (statistical data treatment; discussion on the in situ measurement of Fe2O3 and Na2O) of the current version of the paper.

**Response: Thank you very much for your suggestions on this manuscript. We will answer the questions raised by the editor in the following three points:**

**(1) This manuscript adopts a novel YL-P-3LRX Handheld Laser Induced Breakdown Spectroscopy, which can quickly detect Na elements and achieve high-precision detection of $Na_2O/K_2O$ ratio. Additionally, the rapid detection of $Fe^{2+}$ and $Fe^{3+}$ is indeed very difficult (Chen et al., 2019), which exceeds the detection range of handheld laser-induced breakdown spectroscopy in this manuscript and similar devices. But this factor does not affect the reliability of the quantification criterion for red beds recognition. Ignoring $Fe_2O_3/FeO$ in Equations 1~6, the reliability of the red beds recognition quantification criterion can still reach 95.8% when detecting red beds. In the future, if there are new devices that can quickly detect $Fe^{2+}$ and $Fe^{3+}$, the recognition efficiency of the red beds recognition quantification criterion in this study will be higher.**

2.2 Criterion verification   Lines 181-185

*"After on-site sampling, use a hammer to smash the rock block out of the fresh surface. Then, the fresh surface was analyzed using the YL-P-3LRX Handheld Laser Induced Breakdown Spectroscopy (LIBS, Figure 3) to check whether these elements conform to the basic chemical compositions combination rules of red beds proposed by this study. This device can detect elements such as K, Na, Si, Al, Ca, Mg, Fe, and oxides."*

3.5 Red beds identification quantization criterion verification    Lines 372-386

*"The chemical composition combinations of the 15 selected rocks in this study are shown in Table 5. Study has found that, The rapid detection of $Fe^{2+}$ and $Fe^{3+}$ is very difficult (Chen et al., 2019) and exceeds the detection range of handheld laser-induced breakdown spectroscopy in this manuscript and similar devices. But this factor does not affect the reliability of the quantification criterion for red beds recognition. F1~F5 and F are considered as 6 evaluation indicators, and there are a total of 72 (6 × 12) evaluation indicators for the 12 types of red beds. Among them, 3 evaluation indicators exceed the scope of the quantification criterion for red beds identification (F4 of numbered 7, 9, and 11 red beds with green background in Table 5 is less than the quantification criterion), indicating that the reliability of detecting these 12 types of rocks belonging to the red beds is as high as 95.8%. And for 3 non red beds rocks (limestone, arkose, and mudstone), there are a total of 18 evaluation indicators, of which 13 exceed the scope of the quantification criterion for red beds identification (indicated by blue background), indicating a high reliability of 72.2% in detecting these three types of rocks that do not belong to the red beds. Therefore, this study proposes a quantitative criterion for red beds recognition with high reliability. In the future, if there are new devices that can quickly detect $Fe^{2+}$ and $Fe^{3+}$, the recognition efficiency of the red beds recognition quantification criterion in this study will be higher."*

**(2) Based on the basic statistics of the major element composition of rocks, this study can serve as a preliminary standard for identifying red beds. Through PCA analysis, the principal components of the major element composition of rocks were obtained, which can better quantify the combination of these major elements in rocks and obtain more accurate quantitative criterion for identifying red beds.**

[revised manuscript text omitted]

**(3) This manuscript focuses on the study of the basic chemical composition combinations and quantitative criterion of red beds in the field of geological hazards. According to previous research, geological hazards are to some extent related to the mineral content of red beds, and the main components of these minerals are major elements such as Si, Al, Na, K, Fe, Ca, and Mg. Therefore, this manuscript conducts research on these major elements. Trace elements or stable isotopes can also reliably identify red layers, but they are not the main focus of this manuscript and will be studied separately as our research direction in the future.**

2.1 Data collection      Lines 159-167

*"Studies have found that rock disasters are related to the content of minerals such as quartz, clay minerals, hematite, calcite, dolomite, feldspar, etc., and these mineral contents are also closely related to the combination of major elements or oxides (Table 1), for example, $SiO_2$ and $Al_2O_3$ (used to study the relative content relationship between quartz and clay minerals) (Hong et al., 2009), $Fe_2O_3$ and FeO (used to study the high content characteristics of hematite) (Hu et al., 2006), CaO and MgO (used to study the content relationship of potassium feldspar, calcite, and dolomite) (Han et al., 2023), $Na_2O$ and $K_2O$ (Qiao et al., 2017). Therefore, this study on the basic chemical composition combination rules and quantitative criterion of the red beds only involves the major elements mentioned above, and does not involve the analysis of trace elements or other stable isotopes."*

Minor remarks

Linz 143 and Figure 2 « Mudstone » instead of « Mudtone »

**Response: Modified "Mudtone" to "Mudstone" in manuscript**

Table 1 and Table 2 : add to the caption "from literature data"

**Response: Added "from literature data" to the caption of Tables 1 and 2**

Table 3 : red beds need to be identified in the caption or in the table and discriminated with respect to the other rocks.

**Response: The red beds has been distinguished from other types of rocks in Table 5**